A deep crowd density classification model for Hajj pilgrimage using fully convolutional neural network

Bhuiyan Md Roman 1 romanbhuiyanpv@gmail.com
Abdullah Junaidi 1
Hashim Noramiza 1
http://orcid.org/0000-0003-2625-2348 Al Farid Fahmid 1
Ahsanul Haque Mohammad 2
Uddin Jia 3
http://orcid.org/0000-0001-9060-0250 Mohd Isa Wan Noorshahida 1
Husen Mohd Nizam 4
Abdullah Norra 5
1 Faculty of Computing and Informatics, Multimedia University , Cyberjaya, Selengor , Malaysia
2 Data Scientist and Machine Learning Developer, Aalborg University , Aalborg, Aalborg , Denmark
3 Technology Studies Department, Woosong University , Daejeon , South Korea
4 Information Technology, Malaysian Institute of Information Technology Universiti Kuala Lumpur , Kuala Lumpur , Malaysia
5 Computer Science, WSA Venture Australia (M) Sdn Bhd , Cyberjaya , Malaysia
Ashraf Imran
Electronic publication date: 2022 Mar 25
Publication date: 2022
Volume: 8
Electronic Location ID: e895
Received 2021 Oct 11; Accepted 2022 Jan 26
Copyright: © 2022 Bhuiyan et al.
Copyright year: 2022
Copyright holder: Bhuiyan et al.
License: This is an open access article distributed under the terms of the Creative Commons Attribution License, which permits unrestricted use, distribution, reproduction and adaptation in any medium and for any purpose provided that it is properly attributed. For attribution, the original author(s), title, publication source (PeerJ Computer Science) and either DOI or URL of the article must be cited.
License URL: https://creativecommons.org/licenses/by/4.0/

Keywords: Crowd analysis, Crowd density classification, Fully convolutional neural network (FCNN), Hajj crowd dataset

Funding: FRDGS MMUE/210030 This research is supported by Multimedia University from the grant FRDGS grant (Grant No. MMUE/210030). The funders had no role in study design, data collection and analysis, decision to publish, or preparation of the manuscript.

==============================
This research enhances crowd analysis by focusing on excessive crowd analysis and crowd density predictions for Hajj and Umrah pilgrimages. Crowd analysis usually analyzes the number of objects within an image or a frame in the videos and is regularly solved by estimating the density generated from the object location annotations. However, it suffers from low accuracy when the crowd is far away from the surveillance camera. This research proposes an approach to overcome the problem of estimating crowd density taken by a surveillance camera at a distance. The proposed approach employs a fully convolutional neural network (FCNN)-based method to monitor crowd analysis, especially for the classification of crowd density. This study aims to address the current technological challenges faced in video analysis in a scenario where the movement of large numbers of pilgrims with densities ranging between 7 and 8 per square meter. To address this challenge, this study aims to develop a new dataset based on the Hajj pilgrimage scenario. To validate the proposed method, the proposed model is compared with existing models using existing datasets. The proposed FCNN based method achieved a final accuracy of 100%, 98%, and 98.16% on the proposed dataset, the UCSD dataset, and the JHU-CROWD dataset, respectively. Additionally, The ResNet based method obtained final accuracy of 97%, 89%, and 97% for the proposed dataset, UCSD dataset, and JHU-CROWD dataset, respectively. The proposed Hajj-Crowd-2021 crowd analysis dataset and the model outperformed the other state-of-the-art datasets and models in most cases.

Introduction

There is considerable interest among the scientific community regarding hajj crowd evaluation, especially for pedestrians (Khan, 2015; Ahmad et al., 2018; Ullah, Cheikh & Imran, 2016; Khan et al., 2017). In events such as Hajj, sports, markets, concerts, and festivals, wherein a large number of people gather in a confined space, it is difficult to fully analyze these situations (Khan, Rinner & Cavallaro, 2015; Khan et al., 2017; Saqib et al., 2017a; Khan, 2015; Arif, Daud & Basalamah, 2013). An efficient surveillance system should be implemented to ensure the security and safety of the participants. Despite a great deal of exploration, the regular occurrence of crowd tragedies remains (Ullah et al., 2015; Shuaibu et al., 2015; Khan, Vizzari & Bandini, 2016). In addition, hajj crowd analysis is one of the most important and difficult tasks of video monitoring. The most important use of crowd analysis is to calculate the density of a crowd (Ravanbakhsh et al., 2018; Khan et al., 2016; Saqib et al., 2017b; Wang et al., 2018; Wang et al., 2014; Khan, Crociani & Vizzari, 2014).

One of the requests that has garnered much attention from the scientific community is the calculation of crowd density (Ullah, Ullah & Conci, 2014; Saqib, Khan & Blumenstein, 2017). The density of the crowd in public assembly is necessary to provide useful information to prevent overcrowding, which can lead to a higher risk of stampede. Acknowledging the importance of estimating the density of crowds, numerous attempts have been numerous attempts to overcome this problem by utilizing efficient algorithms (Sabokrou et al., 2018; Ramos et al., 2018; Rota et al., 2013; Ullah & Conci, 2012). This leads researchers to conduct studies and review various crowd density estimation approaches (Zhang et al., 2016b). Researchers have stated that the most stable and effective method of estimating multiple densities in comparison to detection-based methods is texture-based analysis (Ullah & Conci, 2012; Ullah et al., 2010; Khan & Ullah, 2010; Uzair et al., 2009; Khan, Ullah & Hussain, 2013; Saqib, Khan & Blumenstein, 2016; Saleh, Suandi & Ibrahim, 2015).

This study seeks to enhance the categorization of hajj pilgrims based on crowd density. An FCNN-based framework for crowd analysis is presented in this study.

Crowd analysis is inherently a multidisciplinary topic, including scientists, psychologists, biologists, public security, and computer vision experts. Computer vision has grown in relevance in the field of deep learning in recent years. The fully convolutional neural network (FCNN), a profound learning model of grid styling data such as images, is one of the most advanced deep learning models. This technique has the advantage of using seizures during neuronal development and image classification. The FCNN algorithm relies heavily on convolutional, polling, and fully connected layers, as shown in Fig. 1. The convolutional layer is used to represent the image and compute the function mapping. In FCNN, a convolutional layer, which consists of a series of mathematical operations, plays an important role. A polling layer was added after each layer to reduce the resolution of the function mapping. A pool layer is typically used with a sampling approach to reduce the spatial dimension to detect and remove parameters with minimum distortion and function map modifications. Following polling layer sampling, features produced from the convolution layers and such characteristics were created. A totally linked layer of substrates “flattens” the networks used for the input of the next layer. It also contains neurons that are tightly connected to other neurons in two adjacent layers.

Figure 1 Input fully convolutional neural networks output.

The major contributions in this paper included: A fully convolutional neural network (FCNN) was introduced for crowd analysis and estimation of crowd density. We first extracted the frame from the video to estimate crowd density. We sent a full set of images for training and testing and implemented the entire CNN.

Created a deep learning architecture to capture spatial features to automatically evaluate crowd density and classify crowds.

Built a new dataset based on Hajj pilgrimages. We created a new dataset in this research because nobody in the modern world has made this dataset related to Hajj crowds. In State-of-the-art there are few well known crowd datasets worldwide, such as ShanghaiTech, UCSD pedestrians, UCF-CC-50, Mall, WorldExpo, JHU-CROWD, and NWPU-Crowd.

The remainder of this paper is arranged as follows: “Related Work” presents the related work, “Proposed Method” expounds on the proposed method, “Performance Evaluation and Result Analysis” presents the experiment and result discussion, and “Conclusion” presents the conclusion.

Related Work

An early version of crowd analysis has shown that the addition of the Hajj crowd density classification to the crowd analysis system improves its robustness. For example, the study by Hu et al. (2016) a crowd surveillance approach that includes both behavior detection and crowd scene occupation remains to be established. The advantages of multi-task learning have been demonstrated by face analysis (Yan et al., 2015), head-pose prediction (Pan et al., 2016), and voice recognition (Seltzer & Droppo, 2013). The next discussion concentrates on what has been done in each area of crowd analysis.

Crowd analysis

Crowd analysis algorithms are designed to provide an accurate estimate of the real number of people in a crowded image. Crowd analytics has reached a new level of sophistication due to the availability of high-level, high-variable crowd analytics such as UCF CC 50 and the advent of deep network technologies such as convolutional neural networks (Seltzer & Droppo, 2013; Idrees et al., 2013). While most recent methods are used to map pixel values to a single figure (Marsden et al., 2016; Zhang et al., 2015) directly in crowd analysis, pixel-based heat map analysis has shown an increase in the efficiency of crowd analysis for complex, highly congested scenes (Zhang et al., 2016b).

Crowd density level estimation

The degree of congestion in a crowded scene is known as the degree of crowd density. The aspect of a crowded scene usually means a discrete (0–N) value or an ongoing value (0.0–1.0). Texture analysis by Hu et al. (2016) used functionality to continuously produce an estimated density level. For the classification of discrete density levels, Fu et al. (2015) utilized a deep convolutional neural network. This role mainly involves the task of uncertainty associated with a given density level estimate. It is not possible to set density labels and their unique definitions in datasets using a universal framework. The most straightforward scheme is that of discrete labels in density levels explicitly inferred from real multitude analysis values, leading to a distribution of the histogram style with the smallest subjectivity and human error.

Fully convolutional networks

Fully convolutional networks (FCNs) are a CNN variant that provides a proportionally sized map output rather than a mark of classification or regression of the given picture. Two examples of functions used by FCNs are semantic segmentation (Fu et al., 2015) and saliency prediction (Shekkizhar & Lababidi, 2017). In the FCN training, Zhang et al. (2016b) converted a multitude-density heat map of an image of a crowded scene, which can count exactly even in tough scenes. One of the main features of completely convolutional networks, which makes the system especially appropriate for crowd analysis, is the use of an input of variable size that prevents the model from losing information and visual distortions typical of image down sampling and reforming.

The review of contemporary Convolutional Neural Network (CNN)-based algorithms that have proven considerable gains over prior methods that depend heavily on hand-crafted representations. They address the advantages and disadvantages of current CNN-based techniques and highlight prospective research areas in this rapidly expanding subject (Sindagi & Patel, 2018). Additionally, few research works begin with a quick overview of pioneering techniques that use hand-crafted representations before delving into depth and newly released datasets (Al Farid, Hashim & Abdullah, 2019a, 2019b).

The top three performances in their crowd analysis datasets were examined for their advantages and disadvantages based on the assessment measures. They anticipate that this approach will enable them to make realistic conclusions and predictions about the future development of crowd counting while also providing possible solutions for the object counting issue in other domains. They compared and tested the density maps and prediction outcomes of many popular methods using the NWPU dataset’s validation set. Meanwhile, tools for creating and evaluating density maps are included (Gao et al., 2020).

This is mostly due to the fact that Hajj is an absolutely unique event that includes hundreds of thousands of Muslims congregating in a confined space. This article proposes a method based on convolutional neural networks (CNNs) for doing multiplicity analysis, namely crowd counting. Additionally, it presents a novel method for Hajj and Umrah applications. They addressed this issue by creating a new dataset centered on the Hajj pilgrimage scenario (Bhuiyan et al., 2021).

Proposed Method

DEEP-CNN features

A deep CNN can create powerful texture features for each frame without human supervision, unlike handcrafted features, which are susceptible to light fluctuations and noise. Pre-trained CNN models such as VGG19 (Simonyan & Zisserman, 2014), GoogleNet (Szegedy et al., 2015), inceptionv3 (Szegedy et al., 2016), and ResNet101 (He et al., 2016) were evaluated for feature extraction in this research. Every layer is followed by a corrected linear unit layer in VGG19 (Simonyan & Zisserman, 2014). The total number of layers is 50. Rather than processing linearly, GoogleNet’s design (Szegedy et al., 2016) uses several routes, each having 22 weight levels. GoogleNet’s “inception module,” which performs the concurrent processing of multiple convolution kernels, is the building block of the software. While Inceptionv3 (Szegedy et al., 2016) has fewer parameters, it is very fast. As a result, it has the potential to enable a level of complexity comparable to VGGNet, but with deeper layers. Increases in the VGG19, GoogleNet, and inceptionv3 models’ depth lead to saturation and a decline in inaccuracy. On the other hand, ResNet uses skip connections or direct input from one layer to another layer (called identity mapping). ResNet’s skip connections improve the speed of CNNs with a lot of layers thanks to their use. ResNet may also help with the disappearing gradient issue. He et al. (2016) has more information. Due to the strong symbolic capabilities of deep residual networks, ResNet has recently improved the performance of many computer vision applications such as semantic segmentation, object recognition, and image classification.

Python provides access to all of the pre-trained models used in our research. These models were first developed to categorize 224 × 224-pixel images. They may, however, be used to extract features from pictures of any size by using feature extractors.

Crowd density classification process

The crowd density classification process, which consists of three steps, is shown below in Fig. 2. We first performed full image labeling, image data training, and image data testing. In the following section, we describe the details.

Figure 2 Crowd density classification process.

Crowd density process (IMAGE LABELLING)

The crowd density image labeling process followed by manual checking using the Sam counting method based on CNN shows Fig. 3. In this process, the dataset is an image. First, we collected images based on our five classes from YouTube using video capture software. Second, we performed the image selection and validation. For image selection and validation, we applied this rule. If the kaaba is occupied in the middle and people do tawaf on the circle, then we can consider the image to be selected for the labeling process. As an example, based on this rule, we selected the first image shown in Fig. 3. As for the second image, kaaba is not positioned in the center of the image, which is why it cannot be considered for labeling. In the third image, kaaba is in the center of the image; however, the image is zoomed out. Hence, we did not consider this image as one of our datasets. Subsequently, we used the Sam-counting method based on a CNN. In fact, we did not include people. We employed people counting just to aid the process of labeling for density of the five different classes (very low, low, medium, high, and very high). After completing the above steps, we performed manual checking to determine whether the mapping of labels and classes was correct.

Figure 3 Image labeling followed by manual checking on the labeling.

Hajj crowd dataset

This section discusses the proposed HAJJ-Crowd dataset from three perspectives: data capture and specification definition, UCSD, JHU-CROWD and method of annotation.

Data capture and specification definition

HAJJ-crowd dataset were collected from the YouTube live telecast in Mecca Hajj from 2015 to 2019. Accordingly, in some populations surrounding Kaaba (Tawaf area), 27,000 images and 25 video sequences were recorded, including some typical crowd scenes, including touching the black stone in the Kaaba area. We collected images based on five classes, with each class consisting of 5,400 images. Figure 4 shows an example of a five-class dataset.

Figure 4 Example of five classes dataset.

UCSD

Cameras on the sidewalk at UCSD gathered the first crowd-analysis dataset (Chan, Liang & Vasconcelos, 2008). A total of 2,500 frames with a 238 × 158 aspect ratio were used, and every five frames, the ground truth annotations of each pedestrian were added. Linear interpolation was used to generate labels for the remaining frames. Each frame maintains the same viewpoint because it is gathered from the same location. The UCSD dataset did not divided the data category wise. For our experiment we have divided into five classes. The classes are Very Low, Low, Medium, High, and Very High.

JHU-CROWD

JHU-CROWD (Sindagi, Yasarla & Patel, 2019) is one of the largest crowd analysis datasets in recent years. It contains 4,250 images with 330,165 annotations. JHU-CROWD, images are chosen at random from the Internet, busy street. These various scene types and densities are combined to produce a difficult dataset that can be used by researchers. Therefore, the training and test sets tended to be low-density. As a result, many CNN-based networks face new problems and possibilities owing to scale shifts and viewpoint distortions. The JHU-CROWD dataset did not divided the data category wise. For our experiment we have divided into five classes. The classes are Very Low, Low, Medium, High, and Very High.

Method of annotation (Tools)

The annotation tool was developed based on Python and open-cv for easy annotation in Hajj crowd photos based on five classes. The method supports hot encoding label forms and labels the image name based on the threshold. Each image was labeled using the same method.

Implementation

Crowd density process (TRAINING)

Figure 5 shows the training process for crowd density. For the training process, we used an accurate density-labeled image from the previous stage using fully convolutional neural networks (FCNNs) to classify the five classes for training.

Figure 5 Crowd analysis density training process using FCNNs.

Crowd density process (TESTING)

Figure 6 illustrates the crowd analysis density testing process using FCNNs. First, we prepared a new test image dataset. We then passed the full set of images for testing. Second, we tested five classes using the FCNNs. Finally, we obtained classification results for the five classes.

Figure 6 Crowd analysis density testing process using FCNN.

Dataset comparison

According to their findings, the diversity of the dataset makes it impossible for crowd analysis networks to acquire valuable and distinguishing characteristics that are absent or disregarded in the prior datasets, which is our finding. (1) The data of various scene characteristics (density level and brightness) have a considerable effect on each other, and (2) there are numerous erroneous estimates of negative samples. Therefore, there is a growing interest in finding a way to resolve these two issues. In addition, for the localization task, we designed a suitable measure and provided basic baseline models to help start. As a result, we believe that the suggested large-scale dataset would encourage the use of crowd analysis and localization in reality and draw greater attention to addressing the aforementioned issues. Table 1 presents a dataset comparison with other public datasets. Our Hajj-Crowd dataset is based on five classes. The five classes are: very low, low, medium, high, and very high. For our experiment, we used another two datasets, the UCSD and JHU-CROWD datasets. But in the UCSD and JHU-CROWD datasets, they were never divided into different classes. For our evaluation, we have divided five classes manually.

Table 1 Comparison of eight real world dataset.

Dataset	Number of image	Resolutions	Extreme congestion	
UCSD (Chan, Liang & Vasconcelos, 2008)	2,000	158 × 238	No	
UCF-CC-50 (Idrees et al., 2013)	50	2,101 × 2,888	No	
Mall (Chen et al., 2012)	2,000	480 × 640	No	
WorldExpo’10 (Zhang et al., 2016a)	3,980	576 × 720	No	
ShanghaiTech Part A (Zhang et al., 2016b)	482	589 × 868	Yes	
ShanghaiTech Part B (Zhang et al., 2016b)	716	768 × 1,024	Yes	
JHU-CROWD (Sindagi, Yasarla & Patel, 2019)	4,250	1,450 × 900	Yes	
NWPU-Crowd (Wang et al., 2020)	5,109	2,311 × 3,383	Yes	
PROPOSED HAJJ-CROWD DATASET	27,000	1,914 × 922	Yes	

Network for modelling

Consequently, the CNN is implemented as a sequential network. As a result of the convolution operations, a 2D convolution layer is created, which eventually leads to the development of a convolution KERNEL. The following formula was used to compute the subsequent feature map values:

(1) G[m,n]=(f∗h)[m⋅n]=∑j∑kh[j,k]f[m−j,n−k]

The value of f is equal to that of the input image, and the value of h is equal to that of KERNEL. These are the indices of the rows and columns of the final result matrix. We use the filter that we applied to the selected pixel. Then, we take the kernel values for each color and multiply them with their corresponding image values. In summary, the results are output to the feature map. Calculating the dimensions of the output matrix, remembering to account for padding and stride involves finding the

(2) nout=[n+2p−fs+1]

where n is the picture size, f is the filter dimension, p is the padding, and s is the stride. A tensor of the outputs is provided by several layers using numerous input units.

The pooling procedure was performed using MaxPooling 2D software. The max-pooling approach is a sample-based discretisation technique. The goal is to reduce the number of dimensions a dataset possesses to provide a visualization or a hidden layer of data. Assuming that certain features are in the sub-regions and are binned, the positions of such features in the input representation may be estimated. After splitting the data into training and testing sets, data were assessed. We chose 27,000 images for this experiment. In training, twenty one thousand six hundred (21,600) images are used, whereas in testing, 5,400 (20%) images are employed. Before classifier initialization, the training image for the CNN model is produced. After building the model, the first convolution layer is added and initialized as an input layer to the fully connected network that is responsible for the final output layer. When optimizing using the Adam algorithm with a learning ratio of 0.001, we utilized the Adam optimizer with a learning ratio of 0.001. In addition to the training data, test data, and parameters for the number of training steps, the model is also interested in the other three components: training data, test data, and parameters for the number of training steps.

Performance Evaluation and Result Analysis

Experimental setup

The processing of high-resolution images in a fully connected network (e.g., 1,914 × 922 pixels) presents a range of challenges and constraints, particularly with regard to the use of GPU memories. Only certain kernels and layers can have our FCNN convolutional (i.e., model capacity). Therefore, we aim to create the best possible FCNN architecture to process images such as those in a UCSD dataset with the highest possible resolution. We used an NVidia GTX 1660Ti 6 GB RAM 16 GB card. Finally, we utilized python3 in conjunction with deep learning programs such as open-cv2, NumPy, SciPy, matplotlib, TensorFlow GPU, CUDA, Keras, and other similar tools.

Matrix evaluation

The proposed Hajj-Crowd framework’s performance may be verified using the following performance criteria: 1. Precision (Goutte & Gaussier, 2005), 2. Recall (Flach & Kull, 2015), 3. F1 score (Luque et al., 2019), 4. Final accuracy, 5. Confusion metrics (Van der Maaten & Hinton, 2008) and 6. Obtain graph, which illustrates the separability of classes. Precision, Recall, F1 score can be achieved the result by the following equation.

(3) Accuracy=CorrectPredictionCorrectPrediction + IncorrectPrediction

(4) Accuracy=TruePositive + TruenegativeTruePositive + FalsePositive + Truenegative + Falsenegative

(5) Recall=TruePositiveTruePositive + FalseNegative

(6) Precision=TruePositiveTruePositive + FalsePositive

(7) Flscore=2∗Recall*PrecisionRecall + Precision

The terms TP, TN, FN, and FP in Eqs. (3)–(7) denote true positive, true negative, false negative, and false positive, respectively. While evaluating the suggested Hajj-Crowd output, the confusion matrix provides a true overview of the actual vs. projected output and illustrates the performance’s clarity. All the metrics result added on Experiment 1 and Experiment 2.

Experiment 1 (FCNN)

The Hajj crowd dataset is a large-scale crowd density dataset. It includes 21,600 training images and 5,400 test images with the same resolution (1,914 × 922). The proposed method outperforms the state-of-the-art method in the context of a new dataset (Name HAJJ-Crowd dataset), which achieved a remarkable result 100%. For this experiment we have used another two datasets. The datasets are UCSD and JHU-CROWD dataset. The UCSD dataset and JHU-CROWD dataset contain total 2,500 data and each classes contain 500 data and 4,000 and each classes 800. For training we have used 80% and testing 20% respectively. All datasets we have divided into five folds. Figure 7A shows a graph based on the results of the five classes and Fig. 8 shows the Confusion Matrices for the test dataset.

Figure 7 Five classes graphical presentation results.

Figure 8 Confusion matrices for the test dataset, Van der Maaten & Hinton (2008) (A) Hajj-Crowd dataset, (B) UCSD dataset, and (C) JHU-CROWD dataset.

Figure 7B clearly shows that, from 0 to 20 epochs, there is no considerable change in the accuracy, whereas from 20 to 40 epochs, we observed no loss of data. However, for 40 to 100 epochs, the accuracy continued to increase. Finally, the accuracy for 100 epochs was 100%. We have clearly seen that from 0 to 20 epochs there is slowly taking place loss of data, whereas from 20 to 40 and 40 to 100 epochs, there is a rapid loss of data. Finally, the data loss at 100 epochs is 0.01437. On the other hand, Fig. 7B shows the train accuracy and Val-accuracy; from 0 to 20 epochs, there is a slow change in the val-accuracy and at the same time no considerable change in the train accuracy. However, for 20 to 40 epochs and 40 to 100 epochs, it slowly changed with the Train accuracy as well as 20 to 40 and 40 to 100 epochs, with no considerable change in the val-accuracy. Finally, the train accuracy was 0.01437 and the val-accuracy was 1.0000. Figure 7B shows that when the epoch was 0 to 20 epochs, the data loss was 0.35. Subsequently, when the number of epochs increased, the error increased. After completing 100 epochs, we observed that the data loss was slightly high at 0.37. In the val-Loss, we observed that when the epoch was 0 to 20, the data loss was 0.02550. Subsequently, when the number of epochs increases, the data loss decreases. After completing 100 epochs, we observed that the data loss was slightly high at 0.01437.

Each of these comparison tests used the exact experimental dataset. For experiment 1, the FCNN technique obtained a final accuracy of 100%, 98%, and 98.16% on the proposed dataset, the UCSD dataset, and the JHU-CROWD dataset, respectively. The proposed method’s average microprecision, microrecall, and microF1 score are shown in Tables 2–5. All of these assessment matrices were generated using the procedures described in Sokolova & Lapalme (2009). The suggested method’s average microprecision, microrecall, and microF1 score are 100%, 100%, and 100% successively for the proposed dataset, compared to 97%, 97%, 97%, and 95%, 95%, and 95% for the UCSD and JHU-CROWD datasets. All of these results indicate that the developed framework and proposed dataset significantly outperform the two state-of-the-art datasets.

Table 2 Five classes classification report using FCNN model with proposed Hajj-Crowd dataset.

Class	Precision	Recall	F1-score	Support	
VLOW	1.00	1.00	1.00	1,080	
LOW	1.00	1.00	1.00	1,080	
MEDIUM	1.00	1.00	1.00	1,080	
HIGH	1.00	1.00	1.00	1,080	
VHIGH	1.00	1.00	1.00	1,080	
micro avg	1.00	1.00	1.00	5,400	
macro avg	1.00	1.00	1.00	5,400	

Table 3 Three classes classification report using FCNN model with proposed Hajj-Crowd dataset.

Class	Precision	Recall	F1-score	Support	
HIGH	1.00	1.00	1.00	1,080	
VLOW	1.00	1.00	1.00	1,080	
LOW	1.00	1.00	1.00	1,080	
micro avg	1.00	1.00	1.00	3,240	
macro avg	0.60	0.60	0.60	3,240	

Table 4 Five classes classification report using FCNN model with UCSD dataset.

Class	Precision	Recall	F1-score	Support	
VLOW	1.00	1.00	1.00	100	
LOW	0.88	1.00	0.93	100	
MEDIUM	1.00	0.86	0.92	100	
HIGH	1.00	1.00	1.00	100	
VHIGH	1.00	1.00	1.00	100	
micro avg	0.97	0.97	0.97	500	
macro avg	0.98	0.97	0.97	500	

Table 5 Five classes classification report using FCNN model with JHU-CROWD dataset.

Class	Precision	Recall	F1-score	Support	
VLOW	1.00	1.00	1.00	160	
LOW	0.81	1.00	0.89	160	
MEDIUM	1.00	0.76	0.86	160	
HIGH	1.00	1.00	1.00	160	
VHIGH	1.00	1.00	1.00	160	
micro avg	0.95	0.95	0.95	800	
macro avg	0.96	0.95	0.95	800	

Experiment 2 (ResNet)

For each of these comparison tests, the experimental dataset was kept the same. For experiment 2, the ResNet technique obtained final accuracy of 97%, 89%, and 97% for the proposed dataset, UCSD dataset, and JHU-CROWD dataset, respectively. Tables 6–8 illustrate the proposed method’s average microprecision, microrecall, and microF1 score. The formulae from Sokolova & Lapalme (2009) were used to create all of these evaluation matrices. The average micro precision, microrecall, and micro F1 score for the proposed technique is 95%, 95%, and 95% for the proposed dataset, respectively, compared to 90%, 89%, and 90% and 95%, 95%, and 95% for the UCSD and JHU-CROWD datasets. Considering the use of a state-of-the-art dataset that includes JHU-CROWD and UCSD, the proposed FCNN model outperforms in terms of overall accuracy. Additionally, to the best of our knowledge, our proposed dataset is a unique dataset in this domain. Figure 9 shows the graphical results representation with the proposed Hajj-Crowd dataset and two current state-of-the-art dataset.

Figure 9 Graphical result for the training dataset using ResNet method, (A) Hajj-Crowd dataset, (B) UCSD dataset, and (C) JHU-CROWD dataset.

Table 6 Five classes classification report using ResNet model with proposed Hajj-Crowd dataset.

Class	Precision	Recall	F1-score	Support	
VLOW	1.00	1.00	1.00	1,080	
LOW	0.82	1.00	0.88	1,080	
MEDIUM	1.00	1.00	1.00	1,080	
HIGH	1.00	1.00	1.00	1,080	
VHIGH	1.00	1.00	1.00	1,080	
micro avg	0.95	0.95	0.95	5,400	
macro avg	0.96	0.95	0.95	5,400	

Table 7 Five classes classification report using ResNet model with UCSD dataset.

Class	Precision	Recall	F1-score	Support	
VLOW	1.00	1.00	1.00	100	
LOW	1.00	0.46	0.63	100	
MEDIUM	1.00	1.00	1.00	100	
HIGH	1.00	1.00	1.00	100	
VHIGH	0.67	1.00	0.80	100	
micro avg	0.90	0.89	0.90	500	
macro avg	0.93	0.89	0.89	500	

Table 8 Five classes classification report using ResNet model with JHU-CROWD dataset.

Class	Precision	Recall	F1-score	Support	
VLOW	1.00	1.00	1.00	160	
LOW	0.81	1.00	0.89	160	
MEDIUM	1.00	0.76	0.86	160	
HIGH	1.00	1.00	1.00	160	
VHIGH	1.00	1.00	1.00	160	
micro avg	0.95	0.95	0.95	800	
macro avg	0.96	0.95	0.95	800	

Conclusions

This article introduces a novel crowd analysis model that makes use of a fully convolutional neural network (FCNN). The current model of a fully convolutional neural network uses a multi-class structure to address issues in massive crowds. The proposed approach is capable of classifying crowd images into five distinct classes. The results of this experiment demonstrated better performance than the existing state-of-the-art methods. The proposed FCNN-based technique attained a final accuracy of 100% for our own created dataset. Additionally, the proposed dataset was classified with a final accuracy of 97% using the ResNet-based technique. It will help to alert the security personnel before overcrowding happens. In the future, we will focus on overcoming the aforementioned challenges and improving the performance of crowd analysis in the technological world. In addition, we will investigate crowd behavior and pose estimation algorithms in conjunction with temporal information.

Supplemental Information

Supplemental Information 1 High Density Dataset.

Click here for additional data file.

Multimedia University, Cyberjaya, Malaysia fully supported this research.

Additional Information and Declarations

Competing Interests

Author Contributions

Data Availability

Norra Abdullah is employed by WSA Venture Australia (M).

Md Roman Bhuiyan conceived and designed the experiments, performed the experiments, analyzed the data, performed the computation work, prepared figures and/or tables, authored or reviewed drafts of the paper, critical review, and approved the final draft.

Junaidi Abdullah conceived and designed the experiments, performed the experiments, analyzed the data, prepared figures and/or tables, authored or reviewed drafts of the paper, and approved the final draft.

Noramiza Hashim conceived and designed the experiments, analyzed the data, prepared figures and/or tables, authored or reviewed drafts of the paper, and approved the final draft.

Fahmid Al Farid conceived and designed the experiments, authored or reviewed drafts of the paper, and approved the final draft.

Mohammad Ahsanul Haque conceived and designed the experiments, authored or reviewed drafts of the paper, and approved the final draft.

Jia Uddin conceived and designed the experiments, authored or reviewed drafts of the paper, and approved the final draft.

Wan Noorshahida Mohd Isa conceived and designed the experiments, authored or reviewed drafts of the paper, and approved the final draft.

Mohd Nizam Husen conceived and designed the experiments, authored or reviewed drafts of the paper, and approved the final draft.

Norra Abdullah conceived and designed the experiments, authored or reviewed drafts of the paper, and approved the final draft.

The following information was supplied regarding data availability:

The code and dataset are available at GitHub: https://github.com/romanbhuiyan/Code-and-Hajj-Crowd-dataset-2021.

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
