# Peer review of "A deep crowd density classification model for Hajj pilgrimage using fully convolutional neural network"

_PeerJ Computer Science, doi:10.7717/peerj-cs.895_

## Round 0.1 · original submission · Major Revisions

Based on the reviewers' comments, I advise the authors to make "major revisions" and resubmit the revised version.

Reviewer 1 ·

Basic reporting

Authors in their work proposed a new dataset for crowd analysis. In addition to this, proposed a fully convolutional neural network (FCNN)-based method to monitor the crowd. There is merit in the dataset annotation but the manuscript still need a lot of work.

1) Literature references are not sufficient. The related work section needs more work. There is a lot of work done in this field is reviewed in the following papers [1,2].
[1] Sindagi, Vishwanath A., and Vishal M. Patel. "A survey of recent advances in CNN-based single image crowd counting and density estimation." Pattern Recognition Letters 107 (2018): 3-16.
[2] Gao, Guangshuai, et al. "Cnn-based density estimation and crowd counting: A survey." arXiv preprint arXiv:2003.12783 (2020).

2) The English language should be improved to ensure clearly understand your text. Some examples where the language could be improved include lines 80, 93, 157 (what is L denoting here ?), Mathematical notations should be well defined. (For example in 228, What is m and n) – the current phrasing makes comprehension difficult.

Experimental design

These are major issues with the experimental setup.

1. Experiment design and experimental results
Both the proposed method and the benchmark methods are stochastic and the results from multiple independent runs are expected. What is currently reported in the paper is the results from a single run, which is not enough to draw concrete conclusions. Furthermore, multiple runs will be needed to conduct a statistical significance test.

2. Performance metrics
The typical accuracy is inappropriate to be used when the dataset is imbalanced (Shanghai Tech and UCSD) and multiclass (proposed dataset, Shanghai Tech and UCSD). We know for sure that the benchmark datasets in this study fall under that category. Then why the typical accuracy is still used to assess the effectiveness of the experimented methods.

3. Benchmark methods and fairness of the comparisons.
3.1- Benchmark methods
I do not think any of the methods in the experiments was specifically proposed for crowd density classification. There are a number of studies that have proposed similar techniques to the proposed method [1,2], e.g., utilising CNN or other machine learning methods to classify/estimate crowd density. Why none of these was included in the comparisons despite some of such methods being discussed in the related work section.

[1] Gao, Guangshuai, et al. "Cnn-based density estimation and crowd counting: A survey." arXiv preprint arXiv:2003.12783 (2020).

3.2 - Fairness
(a) The proposed method utilises pre-trained models (transfer learning) whereas the benchmark methods are trained from scratch. I do not think this is a fair comparison unless the study is about transfer learning vs conventional learning.
(b) Overall dataset annotation looks fair except there is a chance of having a human bias in 5 classes. As it is very difficult to see the difference in the low and medium (3rd image) and the same is the case with medium and high. In my personal opinion, having three classes (low, medium and high) will be more appropriate than five classes to reduce human bias or error.

Validity of the findings

Experiments 1 and 2 have fundamental issues that need to be resolved before making any valid conclusions.

1. The main issue with stochastic methods is that different results are produced depending on the starting point of the search. In neural networks, the random value generator, more specifically the starting point of the random values generator, initialises the weights; hence, causing the network to start the process from a different point in the search space. Therefore, we must rerun the method multiple times using different seed values “while” keeping everything else untouched/identical.

2. How the other benchmark datasets where category wise evaluation (Mentioned in Table 1) is not available were modified into a classification problem.

3 what are the hyperparameters for all the benchmarks and our proposed model?

4. what are the train and test sizes for other benchmark datasets.

5. Why are recall, f-score etc are not reported?

6. All benchmark models come with pre-trained weights. Which means they are different from each other.
For example, I can use a method that was trained to do anomaly detection and fine-tune it (re-train it) for crowd classification and then compare it to a method that was trained to perform cancer image segmentation in MRIs after re-train it for crowd classification. Can you see the difference between the base models?

Reviewer 2 ·

Basic reporting

no comment

Experimental design

no comment

Validity of the findings

no comment

Additional comments

This article needs important modifications to be suitable for this journal. I suggest major revision for
this paper. The main comments are:

1) In this study, the authors would like to propose a model of deep crowd density classification
for Hajj pilgrimage by using fully convolutional neural network. It can be noticed that the CNN
used in this manuscript has been studied and applied in the previous literatures.
2) The novelty of this paper should be further justified and to establish the contributions to the
new body of knowledge.
3) Abstract section should be improved considering the proposed structure from the journal.
4) In Introduction section, the authors should improve the research background, the review of
significant works in the specific study area, the knowledge gap, the problem statement, and
the novelty of the research.
5) The presentation of the results and conclusions were not enough; it should be highlighted.
6) In the conclusions section, the findings should be explained clearly.
7) The authors should elaborate more on the practical implications of their study, as well as the
limitations of the study, and further research opportunities.
8) The English writing does not influence, in all the paper. There are a lot of grammatical errors
which should be revised by the authors. So, the paper needs a professional English revision.
The author’s guide should be considered by the authors in the writing style in all the paper.

Annotated reviews are not available for download in order to protect the identity of reviewers who chose to remain anonymous.

---

## Round 0.2 · Minor Revisions

In view of the comments from the reviewers, the authors are advised to make 'minor revisions'.

Reviewer 2 ·

Basic reporting

no comment

Experimental design

no comment

Validity of the findings

no comment

Additional comments

This paper needs to consider some suggestions to improve it like: (not taken into account in the first revision)

1- In the conclusions section, the findings should be explained clearly.
2- Conclusions have significantly improved. The authors should elaborate more on the practical implications of their study, as well as the limitations of the study, and further research opportunities.

After the modifications, I suggest to Accept this paper

---

## Author Rebuttal · Round 0.2

09 December 2021

Journal Title: Peer J Computer Science

Manuscript Title: A deep crowd density classification model for Hajj pilgrimage using fully convolutional neural network, CS-2021:10:66465:1:0: NEW

Dear Editor,

Based on your comments on 10 November 2021, we are submitting a revised manuscript titled **A deep crowd density classification model for Hajj pilgrimage using fully convolutional neural network, ID#** CS-2021:10:66465:1:0: NEW**,** for consideration in the *Peer J Computer Science*. I would especially like to thank the reviewers for their thoughtful and constructive comments. These are the most helpful reviewers' comments I have received in 4 years of paper writing. I have included a response letter that outlines how the language editor's comments have been addressed in the revised manuscript. Our responses are highlighted in bold.

We look forward to your positive response.

With Best Regards,

Md Roman Bhuiyan

Multimedia University, Malaysia

Reviewer 1

Basic reporting

Authors in their work proposed a new dataset for crowd analysis. In addition to this, proposed a fully convolutional neural network (FCNN)-based method to monitor the crowd. There is merit in the dataset annotation but the manuscript still need a lot of work.

1) Literature references are not sufficient. The related work section needs more work. There is a lot of work done in this field is reviewed in the following papers [1, 2].

[1] Sindagi, Vishwanath A., and Vishal M. Patel. "A survey of recent advances in CNN-based single image crowd counting and density estimation." Pattern Recognition Letters 107 (2018): 3-16.

**The review of contemporary Convolutional Neural Network (CNN)-based algorithms that have proven considerable gains over prior methods that depend heavily on hand-crafted representations. They address the advantages and disadvantages of current CNN-based techniques and highlight prospective research areas in this rapidly expanding subject (\cite{f62}). Additionally, few research works begin with a quick overview of pioneering techniques that use hand-crafted representations before delving into depth and newly released datasets (\cite{f65, f66}).**

[2] Gao, Guangshuai, et al. "Cnn-based density estimation and crowd counting: A survey." arXiv preprint arXiv:2003.12783 (2020).

**The top three performances in their crowd analysis datasets were examined for their advantages and disadvantages based on the assessment measures. They anticipate that this approach will enable them to make realistic conclusions and predictions about the future development of crowd counting while also providing possible solutions for the object counting issue in other domains. They compared and tested the density maps and prediction outcomes of many popular methods using the NWPU dataset's validation set. Meanwhile, tools for creating and evaluating density maps are included(\cite{f63}).**

**This is mostly due to the fact that Hajj is an absolutely unique event that includes hundreds of thousands of Muslims congregating in a confined space. This article proposes a method based on convolutional neural networks (CNNs) for doing multiplicity analysis, namely crowd counting. Additionally, it presents a novel method for Hajj and Umrah applications. They addressed this issue by creating a new dataset centered on the Hajj pilgrimage scenario.**

2) The English language should be improved to ensure clearly understand your text. Some examples where the language could be improved include lines 80, 93, 157 (what is L denoting here ?), Mathematical notations should be well defined. (For example in 228, What is m and n) – the current phrasing makes comprehension difficult. **Based on the comments from the reviewer, we have updated our draft accordingly.**

Experimental design

These are major issues with the experimental setup.

1. Experiment design and experimental results both the proposed method and the benchmark methods are stochastic and the results from multiple independent runs are expected. What is currently reported in the paper is the results from a single run, which is not enough to draw concrete conclusions. Furthermore, multiple runs will be needed to conduct a statistical significance test.

**Based on the suggestions, we have tried to complete all issues regarding the experimental design. More details on the Experiment 1 (FCNN) and Experiment 2 (ResNet) sections are described in the manuscript.**

2. Performance metrics, The typical accuracy is inappropriate to be used when the dataset is imbalanced (Shanghai Tech and UCSD) and multiclass (proposed dataset, Shanghai Tech and UCSD). We know for sure that the benchmark datasets in this study fall under that category. Then why the typical accuracy is still used to assess the effectiveness of the experimented methods.

**The proposed Hajj-Crowd framework's performance may be verified using the following performance criteria: 1. Precision (C. Goutte and E. Gaussier, 2005), 2. Recall(P. Flach and M. Kull, 2015), 3. F1score (A. Luque, A. Carrasco, A. Mart́ın, and A. de las Heras,2019), 4. Final accuracy, 5. Confusionmetrics (L. van der Maaten and G. Hinton, 2008) and 6. Obtain graph, which illustrates the separability ofclasses. Precision, Recall, F1 score can be achieved the result by the following equation.**

$$\text{Accuracy} = \frac{\text{CorrectPrediction}}{\text{CorrectPrediction} + \text{IncorrectPrediction}} \tag{3}$$

$$\text{Accuracy} = \frac{\text{TruePositive} + \text{Truenegative}}{\text{TruePositive} + \text{FalsePositive} + \text{Truenegative} + \text{Falsenegative}} \tag{4}$$

$$\text{Recall} = \frac{\text{TruePositive}}{\text{TruePositive} + \text{FalseNegative}} \tag{5}$$

$$\text{Precision} = \frac{\text{TruePositive}}{\text{TruePositive} + \text{FalsePositive}} \tag{6}$$

$$\text{F1score} = 2 * \frac{\text{Recall} * \text{Precision}}{\text{Recall} + \text{Precision}} \tag{7}$$

**The terms TP, TN, FN, and FP in Eqs. (3) – (7) denote true positive, true negative, false negative, and false positive, respectively. While evaluating the suggested Hajj-Crowd output, the confusion matrix provides a true overview of the actual vs. projected output and illustrates the performance's clarity. All the metrics result added on Experiment 1 and Experiment 2.**

3. Benchmark methods and fairness of the comparisons.

**Experimental design and experimental results for both the proposed method and the benchmark methods were carried out successfully. Multiple independent runs were also successfully employed in our experiment**

**and updated in the manuscript accordingly. Multiple runs were completed to conduct a statistical significance test through precision, recall, and F1 score.**

3.1- Benchmark methods I do not think any of the methods in the experiments was specifically proposed for crowd density classification. There are a number of studies that have proposed similar techniques to the proposed method [1, 2], e.g., utilising CNN or other machine learning methods to classify/estimate crowd density. Why none of these was included in the comparisons despite some of such methods being discussed in the related work section. Ans: Based on the comments, the manuscript updated considerably.

[1] Gao, Guangshuai, et al. "Cnn-based density estimation and crowd counting: A survey." arXiv preprint arXiv:2003.12783 (2020).
**The given survey paper already cited in our paper at the relevant section.**

3.2 – Fairness (a) The proposed method utilises pre-trained models (transfer learning) whereas the benchmark methods are trained from scratch. I do not think this is a fair comparison unless the study is about transfer learning vs conventional learning.
**Based on the comments, the manuscript updated considerably.**

(b) Overall dataset annotation looks fair except there is a chance of having a human bias in 5 classes. As it is very difficult to see the difference in the low and medium (3rd image) and the same is the case with medium and high. In my personal opinion, having three classes (low, medium and high) will be more appropriate than five classes to reduce human bias or error.
**The comment from the reviewer about three classes (low, medium, and high) is very good and we would like to keep this idea for our future research direction.**

**Validity of the findings**
Experiments 1 and 2 have fundamental issues that need to be resolved before making any valid conclusions.
1. The main issue with stochastic methods is that different results are produced depending on the starting point of the search. In neural networks, the random value generator, more specifically the starting point of the random values generator, initialises the weights; hence, causing the network to start the process from a different point in the search space. Therefore, we must rerun the method multiple times using different seed values "while" keeping everything else untouched/identical.
**Based on the suggestions, we have tried to complete all issues regarding the experimental design. More details on the Experiment 1 (FCNN) and Experiment 2 (ResNet) sections are described in the manuscript.**

2. How the other benchmark datasets where category wise evaluation (Mentioned in Table 1) is not available were modified into a classification problem.
**Since the category wise evaluation on the table causes confusion, we have deleted that column and explained it in the implementation section as below.**

Our Hajj-Crowd dataset is based on five classes. The five classes are: very low, low, medium, high, and very high. For our experiment, we used another two datasets, the UCSD and ShanghaiTech datasets. But in the UCSD and ShanghaiTech datasets, they were never divided into different classes. For our evaluation, we have divided five classes manually.

3 what are the hyper parameters for all the benchmarks and our proposed model?

**Total parameters: 1,617,985**

**Trainable parameters: 1,617,985**

**Number of Epochs**

**Hidden Layers**

**Hidden Units**

**Activations Functions**

4. What are the train and test sizes for other benchmark datasets?

**UCSD has total 2000 image data and each class contain 500. For training 80% and testing 20% were used.**

**ShanghaiTech has a total of 1000 image data sets, and each class contains 200. For training, 80% and testing, 20% were also used.**

5. Why are recall, f-score etc are not reported?

**Based on the reviewer comment we have reported Precision, Recall, F1 score in the manuscript.**

**The proposed Hajj-Crowd framework's performance may be verified using the following performance criteria: 1. Precision (C. Goutte and E. Gaussier, 2005), 2. Recall (P. Flach and M. Kull, 2015), 3. F1score (A. Luque, A. Carrasco, A. Martín, and A. de las Heras, 2019), 4. Final accuracy, 5. Confusion metrics (L. van der Maaten and G. Hinton, 2008) and 6. Obtain graph, which illustrates the separability of classes. Precision, Recall, F1 score can be achieved the result by the following equation.**

$$Accuracy = \frac{CorrectPrediction}{CorrectPrediction + IncorrectPrediction} \quad (3)$$

$$Accuracy = \frac{TruePositive + Truenegative}{TruePositive + FalsePositive + Truenegative + Falsenegative} \quad (4)$$

$$Recall = \frac{TruePositive}{TruePositive + FalseNegative} \quad (5)$$

$$Precision = \frac{TruePositive}{TruePositive + FalsePositive} \quad (6)$$

$$F1score = 2 * \frac{Recall * Precision}{Recall + Precision} \quad (7)$$

**The terms TP, TN, FN, and FP in Eqs. (3) – (7) denote true positive, true negative, false negative, and false positive, respectively. While evaluating the suggested Hajj-Crowd output, the confusion matrix provides a**

**true overview of the actual vs. projected output and illustrates the performance's clarity. All the metrics result added on Experiment 1 and Experiment 2.**

6. All benchmark models come with pre-trained weights. Which means they are different from each other. For example, I can use a method that was trained to do anomaly detection and fine-tune it (re-train it) for crowd classification and then compare it to a method that was trained to perform cancer image segmentation in MRIs after re-train it for crowd classification. Can you see the difference between the base models?

**We performed a model-wise comparison (FCNN and ResNet) as well as a comparison of the proposed dataset and the current two datasets (UCSD and ShanghaiTech).**

Reviewer 2
Additional comments
This article needs important modifications to be suitable for this journal. I suggest major revision for this paper. The main comments are:

1) In this study, the authors would like to propose a model of deep crowd density classification for Hajj pilgrimage by using fully convolutional neural network. It can be noticed that the CNN used in this manuscript has been studied and applied in the previous literatures.

**The CNN used in this manuscript has been studied and applied in the previous literatures, however, we have developed a very unique alternative solution in the domain of Hajj crowd analysis.**

2) The novelty of this paper should be further justified and to establish the contributions to the new body of knowledge.

**We have introduced matrix evaluation based on the advice of another reviewer, which strongly established the contributions.**

3) Abstract section should be improved considering the proposed structure from the journal.

**Abstract section improved considerably.**

4) In Introduction section, the authors should improve the research background, the review of significant works in the specific study area, the knowledge gap, the problem statement, and the novelty of the research.

**We enhanced the research context, the examination of major works in the subject field, the knowledge gap, the issue description, and the research's innovation.**

5) The presentation of the results and conclusions were not enough; it should be highlighted.

**Results and Conclusions improved considerably.**

6) In the conclusions section, the findings should be explained clearly.

 **Conclusions have significantly improved.**

7) The authors should elaborate more on the practical implications of their study, as well as the limitations of the study, and further research opportunities.

**We conducted extensive investigation into the practical implications of our experiment, as well as the study's shortcomings, and more research options were considered.**

8) The English writing does not influence, in all the paper. There are a lot of grammatical errors which should be revised by the authors. So, the paper needs a professional English revision. The author's guide should be considered by the authors in the writing style in all the paper.

**The paper's general writing style improved massively throughout our whole manuscript.**

---

## Round 0.3 · accepted · Accept

Based on the revisions made by authors, the paper is accepted. Congrautlations.